# Laboratory Investigation of Five Inert Dusts of Local Origin as Insecticides against the Colorado Potato Beetle (*Leptinotarsa decemlineata* [Say])

Luka Batistič [1], Tanja Bohinc [1], Aleksander Horvat [2], Iztok Jože Košir [3] and Stanislav Trdan [1,*]

[1] Department of Agronomy, Biotechnical Faculty, University of Ljubljana, Jamnikarjeva 101, SI-1000 Ljubljana, Slovenia; luka.batistic@bf.uni-lj.si (L.B.); tanja.bohinc@bf.uni-lj.si (T.B.)
[2] Scientific Research Centre of Slovenian Academy of Sciences and Arts, Novi Trg 2, SI-1000 Ljubljana, Slovenia; aleksander.horvat@zrc-sazu.si
[3] Slovenian Institute of Hop Research and Brewing, Chemical Analysis and Brewing, Cesta Žalskega Tabora 2, SI-3310 Žalec, Slovenia; iztok.kosir@ihps.si
* Correspondence: stanislav.trdan@bf.uni-lj.si

**Abstract:** The Colorado potato beetle (CPB) is an economically important pest on potatoes, which can cause significant damage if not controlled. Our laboratory trial was conducted to study the efficacy of five types of inert dusts (diatomaceous earth, quartz sand, Norway spruce wood ash, zeolite, and tree of heaven leaf dust) against the CPB. Efficacy was tested using three modes of application (MoA): MoA 1, in which the used inert dusts were applied to both leaves and insects; MoA 2, where just the leaves were pre-dusted; and MoA 3, where only the pests were pre-dusted. All three modes were tested on larvae (L1/L2 and L3/L4) and adults of CPB. Among the inert dusts applied to the adults, the highest efficacy was recorded for the diatomaceous earth treatment (13.3 ± 3.3%) in the first MoA and the wood ash treatment (6.7 ± 3.3%) in the second MoA after 7 days of exposure. Defoliation results for adults were most promising in the *A. altissima* leaf dust treatment in the first and second MoA (45.3 ± 8.2%) after 7 days of exposure. For the old larvae, wood ash was the most promising in regards to efficacy (26.7 ± 7.3%) and defoliation (app. 70.0%) after 7 days of exposure in the third MoA. For the young larvae, the most promising results at the end of the 7-day exposure period for efficacy were obtained from the treatment of diatomaceous earth (65.7 ± 22.9%). Defoliation rates for young larvae were also high, but most promising in the treatment of *A. altissima* leaf dust (41.0 ± 4.2%) in the second MoA. The results showed that the inert dusts were not very effective in controlling the CPB, with the exception of wood ash and diatomaceous earth, which showed some limited control. *A. altissima* leaf dust generated a low defoliation rate, possibly due to an antifeedant effect on the beetles.

**Keywords:** Colorado potato beetle; inert dusts; efficacy; defoliation; wood ash; *A. altissima* leaf dust; alternative control methods; granulometry; geochemistry





## 1. Introduction

The Colorado potato beetle (CPB) (*Leptinotarsa decemlineata* [Say]) (Coleoptera: Chrysomelidae) is one of the most economically important pests of potato plants (*Solanum tuberosum* L.) in the USA and Europe [1,2]. Most damage is caused when the pest is in the adult stage of development, but serious damage is also caused by the larvae [3]. Both stages eat leaves; therefore, they are categorized as foliar pests. Without any method of plant protection, this pest can easily defoliate up to 100 cm$^2$ of leaf area per specimen in its lifetime [2,4]. It can also cause a 100% defoliation rate if potato plants are not treated. This may result in more than a 50% yield loss of potato tubers [2,4]. This pest has two to three generations per year, which represents an even greater concern for effective plant protection [1]. The Colorado potato beetle is usually controlled using insecticides.

In some countries, due to excessive use and improper insecticide rotation, some beetle populations show signs of insecticide resistance [5,6]. Therefore, new management programs and potato production strategies were developed with the intention of diminishing or preventing this recurring problem [7,8]. As we know from other studies, the CPB is capable of gaining resistance to various groups of insecticides [9], which alone demands some form of optimization and development of environmentally acceptable methods for controlling this pest.

Inert dusts, especially those based upon activated silicas, have potential for pest control and are being used as storage protectants in the grain industry [10]. Some testing of the use of inert dusts against agricultural pests, such as the CPB and others, have also been performed in laboratory and field conditions [11–13]. Inert dusts act on insects physically and on the respiratory system, and therefore generally work slower, i.e., through specific contact, unlike conventional insecticides that act as contact poisons, protoplasmic poisons, respiratory poisons, and nerve poisons [10,14]. The mode of action of these substances varies depending on particle size, uniformity and shape, pH and formulation purity [11]. The most common insect death occurs due to desiccation: water loss as the result of the destruction of the insect's cuticle. Some silica agrogels also affect the insect cuticle by absorbing particles of protective wax [14,15]. One of the advantages of inert dusts is their low toxicity to mammals. In the US, for example, all types of diatomaceous earth are recognized as safe by the US Food and Drug Administration. Therefore, they are registered as food additives [16].

Worldwide, the efficacy of inert dust on the CPB has not yet been sufficiently studied. Some examples of their use can be found in studies conducted in the USA [12] and Canada [13]. The research of Becker [12] shows the possibility of using various alternative methods to control the CPB on potatoes. One method is the use of diatomaceous earth as a potentially effective agent against the pest. Wood ash was also used in research as a substance to control the CPB. It has been shown to be efficient against CPB [13] since it serves as a physical barrier and prevents free movement of beetles among potato plants. Thus, with studies providing positive results and others providing weaker results for the use of inert dusts, we were interested in revaluating the efficacy of these substances in laboratory trials.

In Slovenia, the use of inert dusts as protectants has been studied mostly against weevils and other grain pests [17,18]. Studies have shown that Norway spruce ash is very effective in controlling rice weevils in grains [17]. Diatomaceous earth is also very effective, but it must have a high percentage of $SiO_2$ (more than 95%). Other studies that involve CPB in connection with the use of inert dusts are scarce, and they were mainly conducted outdoors in the form of field experiments [19,20]. The results from the laboratory research thus prove an adequate suitability of wood ash for the purpose of CPB management [19]. The aim of this research was to determine whether various types of inert dust can efficiently control the CPB in laboratory trials and if so, to highlight the properties of each one and how it probably affects the CPB.

## 2. Materials and Methods

### 2.1. Inert Dusts and the Colorado Potato Beetle

The investigation was carried out in 2022 in the Laboratory of Phytomedicine of the Chair of Phytomedicine, Agricultural Engineering, Field Crop Production, Grassland and Pasture Management (University of Ljubljana, Biotechnical Faculty, Department of Agronomy) in Ljubljana, Slovenia. The following five types of inert dusts of local origin, plus the positive and negative controls, were tested: diatomaceous earth (local origin from Bela Cerkev, 45.864301, 15.269119), quartz sand (local origin from Puconci, 46.712503, 16.158781; supplier: Kema Puconci, Slovenia), ground leaves of the tree of heaven (*Ailanthus altissima* [Mill.]), wood ash from Norway spruce (obtained from a local farm at Zgornja Lipnica, 46.321906, 14.185115) and zeolite (local origin from Zaloška Gorica, 46.276203, 15.202337, supplier: Montana Ltd., Žalec, Slovenia). For the purpose of the posi-

tive control, we used the insecticide Laser plus (producer: Corteva Agrisciences Bulgaria; supplier: Karsia Dutovlje d.o.o., Slovenia) with its active ingredient spinosad (spinosin A+spinosin D, 48%), and for the negative control, we only used beetles/larvae and untreated fresh potato leaves. All of the abovementioned inert dusts and previously dried and ground leaves of *A. altissima* were obtained from a location in Ljubljana (46.072600, 14.494781). All of the dusts were properly stored in waterproof containers, and these containers were kept in a dry place. Safe storage was also applied for the spinosad-based insecticide. Before each use, we also thoroughly examined and sifted the dusts to enable optimal application and remove possible irregularities and larger particles. Larvae (L1/L2 and L3/L4) and adults of the CPB were collected on potato plants (cv. 'Belmonda') from a specific experimental field of the Biotechnical Faculty in Ljubljana (46.050071, 14.470959), where insecticides were not used at all. Adults and larvae of both generations were used in the laboratory trials. We collected young larvae (L1/L2) in the first half of June and in the first half of August. Older larvae (L3/L4) were collected in the second half of June and in the second half of August. Adults were also collected in the middle of July and in the middle of September. Individuals were handpicked, placed in plastic containers, and transported to the laboratory where they were used for laboratory experiments with inert dusts.

*2.2. Laboratory Bioassay*

The efficacy of the inert dusts was tested with the following three different application modes: (1.) dusted beetles placed on dusted potato leaves; (2.) undusted beetles placed on dusted potato leaves; and (3.) dusted beetles placed on undusted potato leaves. In the first MoA, we aimed to observe the behavior of the CPBs when they were pre-dusted as well as their food source. Our attempt was to mimic the field conditions where the inert dust would be applied, and both the beetles and the plants would directly come into contact with it. In the second MoA, we investigated the efficacy of the agent even if the pest is not directly exposed to it. In real-world scenarios, this could be demonstrated when the pest takes refuge under the plant's leaf structure. The third MoA involved replicating the pest's response when it comes into contact with the substance without it being applied to the plant. This method is challenging to apply in an outdoor setting, and may require using barriers next to the potato plants on the ground, as referred to in a similar article [13]. Therefore, we used separate containers for each of the 7 treatments. Considering 3 repetitions per treatment and the abovementioned application methods (3 in total) of the studied inert dusts, there were 63 plastic containers (measurements: 20 cm × 15 cm × 5 cm) in total, with a volume of 1.5 L each.

For the purpose of the experiment, we used small 1.5 L containers in which we first placed 2 fresh potato leaves. To preserve the freshness of the leaves, we wrapped the ends of their stalks in water-soaked cotton wool and made sure to add water as needed. In each container, we then also added 20 CPB L1/L2 larvae. The same was performed with L3/L4 larvae. For the adult beetles, we collected less, and we used 10 adult unsexed CPBs per container.

Depending on the method of application, the beetles and leaves were also dusted accordingly. We used app. 5 g of inert dust for each treatment. The inert dusts were applied to the leaves by shaking the leaves in a container with the specific inert dust. By weighing the leaves before and after applying the powders and measuring the surface of the leaves, we calculated that an average of 0.10 g (*A. altissima*) to 0.29 g (zeolite) of inert dust was applied to 100 cm$^2$ of leaves, which increased the mass of the powdered leaves from 4.4% (diatomaceous earth) to 13.20% (zeolite) (Table 1).

**Table 1.** Density of the inert dusts and their influence on the increase in leaf mass and the inert dust weight percentage of the total weight of the tested potato leaves (results are based on 10 potato leaf averages for each inert dust).

| Inert Dust | Density (g/L) | Increase in Leaf Mass after Dust Application (g/100 cm$^2$) | Inert Dust Weight Percentage Based on the Weight of the Potato Leaf, after Application (%) |
|---|---|---|---|
| Quartz sand | 1310.98 ± 81.22 | 0.23 ± 0.02 | 10.30 ± 1.22 |
| Zeolite | 989.00 ± 52.89 | 0.29 ± 0.03 | 13.20 ± 0.98 |
| Diatomaceous earth | 495.66 ± 75.02 | 0.13 ± 0.01 | 4.40 ± 0.33 |
| Norway spruce wood ash | 311.25 ± 24.10 | 0.14 ± 0.02 | 6.10 ± 0.26 |
| *A. altissima* leaf dust | 320.37 ± 32.77 | 0.10 ± 0.02 | 4.90 ± 0.19 |

In the third MoA, the beetles were exposed to the dust by shaking them in a glass jar for 1 min, and they were then placed in containers with leaves. In this way, we tried to simulate the foliar application of inert dusts. For the purpose of the negative control, we used untreated leaves and CPB specimens. For the positive control, we applied the prescribed dose (0.4 mL/10 L H$_2$O) of the insecticide Laser Plus with a pump action hand pressure sprayer on the leaves in all three MoAs. The leaves were placed in plastic containers only after the spray had dried on them. This type of testing with similar methodology is widely used in the field of storage pests, most actively with weevils [17,21]. A similar methodological approach can also be observed in laboratory tests with the CPB [13,22].

The containers were placed in an RK-900 CH-type rearing chamber provided by Kambič Laboratory equipment (Semič, Slovenia) with a working capacity of 0.868 m$^3$ (width × height × depth = 1000 × 1400 × 620 mm) [9]. Each treatment was performed in 3 replicates. The containers were kept at a 12-h day and night interval at 20 °C and a relative humidity of 75%. The number of dead individuals was determined at 1, 2, 3, 4, and 7 days after treatment (DAT). At the same time, we also evaluated the level/degree of defoliation on the compound potato leaves by visually assessing the extent of CPB defoliation on days 1, 2, 3, 4, and 7 following the treatment.

*2.3. Geochemical Analysis of Inert Dusts*

The geochemical analyses were performed at Activation Laboratories Ltd., Ancaster, Ontario, Canada. The most aggressive ICP fusion technique employs a lithium metaborate/tetraborate fusion for whole rock analysis. Fusions using a robotic system, which provides a fast fusion of the highest quality. The resulting molten bead is rapidly digested in a weak nitric acid solution. The fusion ensures that the entire sample is dissolved. Samples are prepared and analysed in a batch system. Each batch contains a method reagent blank, certified reference material, and 6% replicates. Samples are mixed with a flux of lithium metaborate and lithium tetraborate and fused in an induction furnace. The molten melt is immediately poured into a solution of 5% nitric acid containing an internal standard, and mixed continuously until completely dissolved. The samples are run for major oxides and selected trace elements on an ICP. Calibration is performed using 14 prepared USGS and CANMET certified reference materials. One of the 14 standards is used during the analysis for every group of ten samples.

Low SiO$_2$ content for analysed diatomaceous earth is due to the sampling factor. The composition of Slovenian diatomaceous sediments varies due to SiO$_2$ and CaO content [23]. In our previous research, we used diatomaceous sediments with high SiO$_2$ content [17,24]. At the sampling location Bela Cerkev, two types of diatomaceous sediments could be found [23]: diatomaceous carbonate siltstones with low SiO$_2$ content (<15%) and high CaO content (>40%), and diatomaceous siltstones with high SiO$_2$ content (>65%) and low CaO content (<1%). The results of our geochemical analysis can be seen in Table 2.

**Table 2.** Results of geochemical analysis of inert dusts used in our survey.

| Chemical Substance/Element | Unit | Diatomaceous Earth | Quartz Sand | Zeolite | Norway Spruce Wood Ash |
|---|---|---|---|---|---|
| $SiO_2$ | % | 10.74 | 96.47 | 65.86 | 13.46 |
| $Al_2O_3$ | % | 2.65 | 0.29 | 13.05 | 4.25 |
| $Fe_2O_3(T)$ | % | 0.89 | 1 | 3.52 | 1.34 |
| MnO | % | 0.362 | 0.014 | 0.054 | 1.801 |
| MgO | % | 5.58 | 0.02 | 1.66 | 4.3 |
| CaO | % | 31.15 | 0.12 | 2.5 | 37.05 |
| $Na_2O$ | % | 0.18 | 0.03 | 2.87 | 0.53 |
| $K_2O$ | % | 10.29 | 0.06 | 2.19 | 7.42 |
| $TiO_2$ | % | 0.173 | 0.029 | 0.395 | 0.218 |
| $P_2O_5$ | % | 1.35 | <0.01 | 0.06 | 2.74 |
| LOI | % | 35.26 | −0.04 | 8.77 | 26.34 |
| Total | % | 98.62 | 97.99 | 100.9 | 99.46 |
| Ba | ppm | 711 | 15 | 770 | 2565 |
| Sr | ppm | 324 | 3 | 590 | 495 |
| Y | ppm | 4 | <1 | 19 | 7 |
| Sc | ppm | 2 | <1 | 6 | 2 |
| Zr | ppm | 12 | 29 | 191 | 8 |
| Be | ppm | <1 | <1 | 2 | <1 |
| V | ppm | 18 | <5 | 38 | 20 |

For *A. altissima*, we used the following procedure: collection of plant material, chemical analysis, preparation of sample extracts, preparation of standard solutions, use of HPLC, and geochemical analysis. Everything was analysed exactly as described in the article of Bohinc et al. [18]. The relative humidity (%) of the analysed sample was around 8.1%, while the sample average essential oil content was 0.01%. For purposes of a more accurate representation of the components, we analysed the sample twice. The results can be seen in Table 3.

**Table 3.** Average values of polyphenols presented as mg/g of *A. altissima* leaf mass for leaves sampled on 15 July 2021.

| Polyphenols | mg/g |
|---|---|
| Weight percentage (%) | $2.93 \pm 0.14$ |
| Catechyin hydrate | $18.89 \pm 0.80$ |
| Naringin | $2.09 \pm 0.09$ |
| Hydroxy coumarin | ND |
| Quercetin | $0.52 \pm 0.30$ |
| Caffeic acid | $0.26 \pm 0.01$ |
| p-coumaric acid | ND |
| Ferulic acid | $0.04 \pm 0.01$ |
| Rutin hydrate | $3.79 \pm 0.16$ |
| Quercitrin | $1.05 \pm 0.04$ |

ND stands for not able to define.

### 2.4. Granulometry

For granulometric analysis, the Fritsch Analysette 22–28 device (manufacturer: Fritsch GmbH, Idar-Oberstein, Germany), located at the Department of Geology, University of Ljubljana, Slovenia was used, which combines a laser granulometer and dynamic image analysis. The size range of measurements with the mentioned equipment varies between 0.01 μm and 2100 μm. Due to the larger amount of the sample, the latter is quartered until a representative sample is obtained for further analysis (approx. 1–2 g). The sample was then poured with distilled water and ultrasonicated for 10 min to break up any agglomerates. This created an even suspension. The suspension was then measured with a laser granulometer. Each individual sample measurement was performed three times to ensure the representativeness of the measured data. From all of the measured curves of

the individual samples, a representative average curve was created, in which all measured fractions and their relative representation were considered. Results from the granulometric analysis of inert dusts can be seen in Table 4.

**Table 4.** Results of granulometric analysis of inert dusts used in analysis.

| Grain Size (µm) | Diatomaceous Earth | Quartz Sand | Zeolite | Norway Spruce Wood Ash |
|---|---|---|---|---|
| Arithmetic mean grain size | 321.64 ± 212.38 µm | 239.58 ± 43.65 µm | 293.56 ± 80.37 µm | 196.43 ± 99.90 µm |

*2.5. Statistical Analysis*

We conducted multifactor analysis of variance (ANOVA) to determine the differences in efficacy (%) and defoliation (%) between the larvae (L1/L2, L3/L4) and adults of the CPB. Before the analysis, each variable was tested for homogeneity of treatment variances. The efficacy of treatment data was corrected according to Abbott's formula and normalized using the arcsine square-root transformation [25]. In the statistical analysis of the results of defoliation, we used the results from the negative control (untreated) as an equivalent factor (treatment) to the other six treatments. Significant differences ($p < 0.05$) between mean values were identified using Student-Newman–Keuls's multiple range test. All statistical analyses were performed using Statgraphics Plus for Windows 4.0 (Statistical Graphics Corp., Manugistics, Inc., Rockville, MD, USA). All data are presented as untransformed means ± SEs.

**3. Results**

*3.1. Efficacy*

3.1.1. General Insight

ANOVA of pooled results indicated a statistically significant effect of the day of exposure (F = 27.67; df = 4; $p < 0.0001$), type of treatment (F = 1800.15; df = 5; $p < 0.0001$), the mode of application (F = 9.58; df = 2; $p < 0.0001$), and the developmental stage of the CPB (F = 42.49; df = 2; $p < 0.0001$) on the average efficacy of treatments in the experiment. The interaction between the day of exposure and the developmental stage (F = 4.09; df = 8; $p < 0.0001$), the type of treatment, the mode of application (F = 4.97; df = 10; $p < 0.0001$), the type of treatment and the developmental stage (F = 14.22; df = 10; $p < 0.0001$), the mode of application and developmental stage (F = 42.49; df = 4; $p < 0.0001$), and the interaction between the type of treatment, the mode of application, and the developmental stage (F = 5.64; df = 20; $p < 0.0001$) all affected the efficacy of treatments on the CPB specimen in the experiment.

ANOVA for young larvae (L1–L2) indicated a statistically significant effect of the day of exposure (F = 9.35; df = 4; $p < 0.0001$), the type of treatment (F = 308.55; df = 5; $p < 0.0001$) and the MoA (F = 13.93; df = 2; $p < 0.0001$). We detected only one significant interaction that had an effect on the efficacy of treatments on the young larvae. The interaction between the type of treatment and the MoA was significant (F = 6.75; df = 10; $p < 0.0001$).

ANOVA of pooled results of old larvae (L3–L4) indicated a statistically significant effect of the days of exposure (F = 24.20; df = 4; $p < 0.0001$), the type of treatment (F = 786.29; df = 5; $p < 0.0001$), and the MoA (F = 14.76; df = 2; $p < 0.0001$). None of the three types of interactions had a significant effect on the efficacy of treatments on the old larvae of the CPB in the experiment. The interaction between the type of treatment and the MoA (F = 3.71; df = 10; $p < 0.0001$) was the only interaction that had any effect on the efficacy of treatments on the old larvae in the experiment.

ANOVA of pooled results of adults indicated a statistically significant effect of the day of exposure (F = 10.00; df = 4; $p < 0.0001$) and the type of treatment (F = 675.74; df = 5; $p < 0.0001$). All three types of interactions affected the efficacy of treatments on the adult CPB specimen in the experiment. The interaction between the type of treatment and the days of exposure (F = 3.45; df = 20; $p < 0.0001$), between the day of exposure and the MoA

(F = 4.66; df = 8; *p* < 0.0001), and between the type of treatment and the MoA (F = 3.71; df = 10; *p* < 0.0001), were examined.

### 3.1.2. Mode of Application 1

ANOVA of pooled results for the mode of application 1 indicated a statistically significant effect of the day of exposure (F = 5.35; df = 4; *p* < 0.0001), type of treatment (F = 299.44; df = 5; *p* < 0.0001), and the developmental stage (F = 27.15; df = 2; *p* < 0.0001). All three affected the efficacy of treatment on the CPB in the experiment. The only interaction that had a statistically significant effect on the efficacy of treatments in MoA 1 was between the type of treatment and the developmental stage (F = 8.23; df = 10; *p* < 0.0001).

On the first day of exposure, we found the lowest average efficacy (17.2 ± 4.8%). Grand means and parentheses are indicated as results, presented in this paragraph and the second paragraphs of Sections 3.1.3 and 3.1.4 in this paper. The significantly highest average efficacy of treatments was recorded at the seventh day of exposure, reaching almost 30% (29.1 ± 5.4%). Regarding the treatment, by far the highest average efficacy on the CPB specimens was recorded in the positive control treatment (98 ± 0.6%). Significantly, the lowest average efficacy among treatments was recorded in the *A. altissima* treatment (1.8 ± 0.8%). As for the developmental stage, the highest average significant efficacy of treatments among all developmental stages was recorded among the young larvae (32 ± 4.1%). Significantly, again, the lowest average efficacy of treatments among the developmental stages of the CPB was recorded among the old larvae (19.1 ± 3.8%).

Based on the analysis, at the first MoA, only treatment type had a significant effect on the overall efficacy on the young larvae (F = 39.11; df = 5; *p* < 0.0001). The interaction between the type of treatment and the days of exposure was not significant. In regards to the treatment type, as shown in Figure 1, the overall efficacy on the young larvae was the highest in the positive control treatment (insecticide) from Day 1 (96.7 ± 1.7%). Among the studied inert dusts, the wood ash treatment showed some effect on efficacy (17.2 ± 10.1%) on the seventh day of exposure. The highest efficacy on the young larvae between the used inert dusts was recorded for the treatments of diatomaceous earth (65.7 ± 22.9%) and zeolite (46.3 ± 27.3) on the seventh day of exposure (Figure 1).

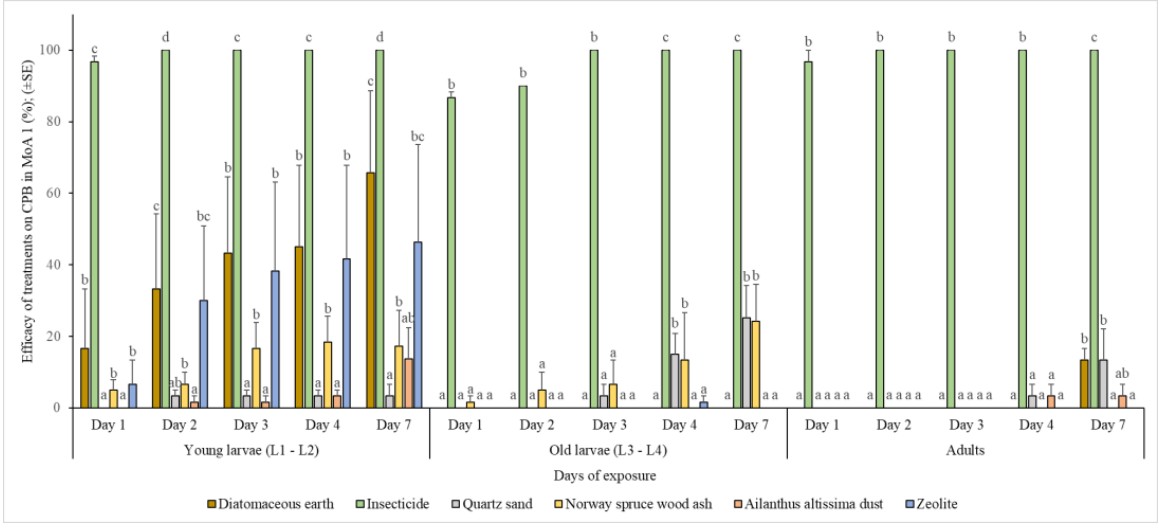

**Figure 1.** Efficacy of treatments (±SE) on CPB in MoA 1 (dusted beetles placed on dusted potato leaves) per day of exposure and developmental stage (different lowercase letters represent significant differences within the same developmental stage between different treatments by day of exposure).

Based on the analysis, we found that at the first MoA, treatment type (F = 219.49; df = 5; $p < 0.0001$) was the only factor with a significant effect on the overall efficacy of treatments on the old larvae. The interaction between treatment type and days of exposure was not significant. In regards to the treatment type, as shown in Figure 1, the overall efficacy on the old larvae was the highest in the positive control treatment (insecticide) from Day 1 (86.7 ± 1.7%) until Day 7 (100 ± 0.0%). In regards to the inert dusts, wood ash had little efficacy (6.7 ± 6.7%) during the first 3 days of exposure. Among inert dusts, the highest efficacy on old larvae was observed with the wood ash (24.2 ± 10.3%) and quartz sand (25.3 ± 9.0%) treatments on the seventh day of exposure (Figure 1).

Based on the analysis of the adults, we found that at the first MoA, there was a significant effect of treatment (F = 1712.93; df = 5; $p < 0.0001$) and day of exposure (F = 125; df = 4; $p < 0.0001$). The interaction between both (F = 68.23; df = 20; $p < 0.0001$) was also significant and affected the overall efficacy of treatments on the adult specimens. In regards to the treatment type, as shown in Figure 1, the overall efficacy on the adult beetles was the highest in the positive control treatment (100 ± 0.0%), where we used an insecticide. For the studied inert dusts, there was little if no effect during the first 4 days of exposure. The highest efficacy on the adult specimens was recorded in the treatments of diatomaceous earth (13.3 ± 3.3%) and quartz sand (13.3 ± 8.8%) on the seventh day of exposure (Figure 1).

### 3.1.3. Mode of Application 2

ANOVA of pooled results for the mode of application 2 indicated a statistically significant effect of the day of exposure (F = 6.46; df = 4; $p < 0.0001$), type of treatment (F = 1319.10; df = 5; $p < 0.0001$) and the developmental stage (F = 9.10; df = 2; $p < 0.0001$). All three affected the efficacy on the CPB. The only interaction that had a statistically significant effect on the efficacy of treatments on the CPB in MoA 2 was between the type of treatment and the developmental stage of the pest (F = 3.09; df = 10; $p < 0.0001$).

On the first day of exposure, we found, significantly, the lowest average efficacy on the CPB specimens (16.6 ± 4.9%). The highest average efficacy of treatments was recorded at the seventh day of exposure (23 ± 5%). Regarding the treatment, the highest average efficacy on the CPB specimens was recorded in the positive control treatment (98.7 ± 0.5%). Significantly, the lowest average efficacy on the CPB specimens was recorded in the *A. altissima* treatment (1.6 ± 0.4%) and the quartz sand treatment (2.4 ± 0.9%). As for the developmental stage, the highest average significant efficacy of treatments among all developmental stages was recorded among the young larvae (21.8 ± 3.8%). Significantly, again, the lowest average efficacy of treatments among the developmental stages of the CPB was recorded among the adults (17.3 ± 3.9%).

Based on the analysis, at the second MoA, only treatment type had a significant effect on the overall efficacy on the young larvae (F = 235.89; df = 5; $p < 0.0001$). In general, we also did not find a significant interaction between the type of treatment and the days of exposure. Considering the results depicted in Figure 2, we can infer that the overall efficacy of the used inert dusts in the second MoA was inferior to the efficacy presented in the first MoA. The most effective treatment was that of the positive control, with a 100% efficacy rate on the first day of exposure (100.0 ± 0.0%). The efficiency of the inert dusts was unsatisfactory. On the seventh day of exposure, the highest significant efficacy on the young larvae between the used inert dusts treatments was caused by diatomaceous earth (18.3 ± 18.3%), wood ash (15.9 ± 3.2%), zeolite (10.4 ± 3.0%), and quartz sand (8.5 ± 4.4%). Overall, the most efficient treatment was the positive control (Figure 2).

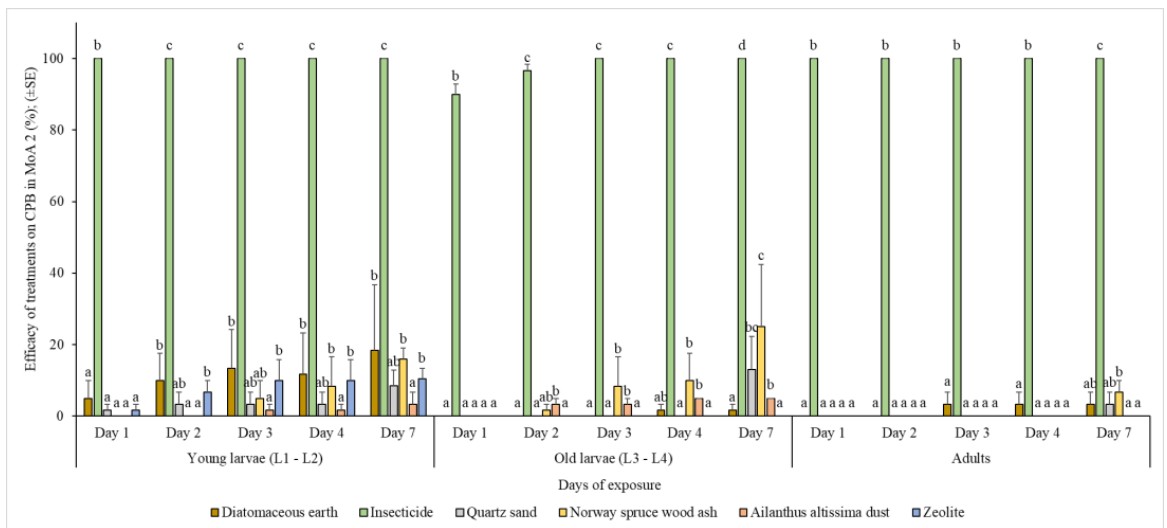

**Figure 2.** Efficacy of treatments (±SE) on CPB in MoA 2 (undusted beetles placed on dusted potato leaves) per day of exposure and developmental stage (different lowercase letters represent significant differences within the same developmental stage between different treatments by day of exposure).

For the second MoA, we found a significant effect of treatment (F = 391.92; df = 5; $p < 0.0001$) but not of the days of exposure on the efficacy of treatments on the old larvae. We also did not find a significant interaction between the type of treatment and the days of exposure. Considering the results shown in Figure 2, we can also deduce that the overall efficacy of the inert dusts tested using the second MoA against the old larvae was similar and comparable to the first MoA. The most effective treatment was that of the positive control, with an almost 100% efficacy rate on the second day of exposure (96.7 ± 1.7%). The efficiency of the inert dusts was unsatisfactory. On the seventh day of exposure, the highest significant efficacy on old larvae between the used inert dusts was observed for wood ash treatment (25.0 ± 17.3%). Overall, the most efficient treatment was the positive control treatment.

For the second MoA, we found a significant effect of the type of treatment (F = 4397.16; df = 5; $p < 0.0001$) but no significant effect of the day of exposure on the average efficacy on adult CPB. We also did not find significant interactions between treatments and days of exposure. Considering the results shown in Figure 1, we can also deduce that the overall efficacy of the used dusts was somewhat similar to the previous MoA. We observed that inert dusts caused no significant efficacy on the adult specimens of CPB in both application modes (MoA 1 and MoA 2). On the seventh day of exposure, significantly higher efficacy was achieved with wood ash treatment (6.6 ± 3.3%) on the adult beetles between the used inert dusts. Overall, the most efficient treatment was the positive control treatment.

### 3.1.4. Mode of Application 3

ANOVA of pooled results for the mode of application 3 indicated a statistically significant effect of the day of exposure (F = 30.85; df = 4; $p < 0.0001$), type of treatment (F = 1013.91; df = 5; $p < 0.0001$) and the developmental stage (F = 21.80; df = 2; $p < 0.0001$) on the efficacy. All of the interactions also had a statistically significant effect on the efficacy of treatments on the CPB in MoA 3. Firstly, the interaction between the day of exposure and the type of treatment (F = 1.82; df = 20; $p < 0.0001$), the day of exposure, and the developmental stage of the pest (F = 7.98; df = 8; $p < 0.0001$), plus the type of treatment and the developmental stage of the pest (F = 13.87; df = 10; $p < 0.0001$).

On the first day of exposure, we found, significantly, the lowest average efficacy of treatments on the CPB specimens (16.8 ± 4.7%). The highest average efficacy of treatments was recorded at the seventh day of exposure (32.1 ± 5%). Regarding the treatment, the highest average efficacy on the CPB specimens was recorded in the positive control treat-

ment (98.2 ± 0.7%). Significantly, the lowest average efficacy on the CPB specimens was recorded in the zeolite treatment (3 ± 1.2%) and the quartz sand treatment (4 ± 1.7%). As for the developmental stage, the highest average significant efficacy of treatments among all developmental stages was recorded among the old larvae (25.1 ± 3.8%) and the young larvae (23.4 ± 3.8%). Significantly, the lowest average efficacy of treatments among the developmental stages of the CPB was recorded among the adults (17.7 ± 3.9%).

Based on the analyses for the third and final MoA (dusted CPB on untreated leaves), there was a significant effect of the type of treatment (F = 283.47; df = 5; *p* < 0.0001) and the days of exposure (F = 13.94; df = 4; *p* < 0.0001) on the average efficacy of treatments on the young larvae in the experiment. The interaction between the type of treatment and the days of exposure was not significant for young larval killing efficacy. From Figure 3, we can deduce that the most effective treatment was that of the positive control (100 ± 0.0%) after the second day of exposure. From the graph, we can also see that the *A. altissima* dust treatment generated better results with the third MoA than with the other application modes, but for the other treatments, the results were poorer than those with the first MoA. Of the inert dusts used, diatomaceous earth (31.7 ± 6.0%), *A. altissima* dust (32.5 ± 7.5%), and quartz sand (26.9 ± 19.6%) treatments generated the most promising results at the end of the seventh day of the exposure period.

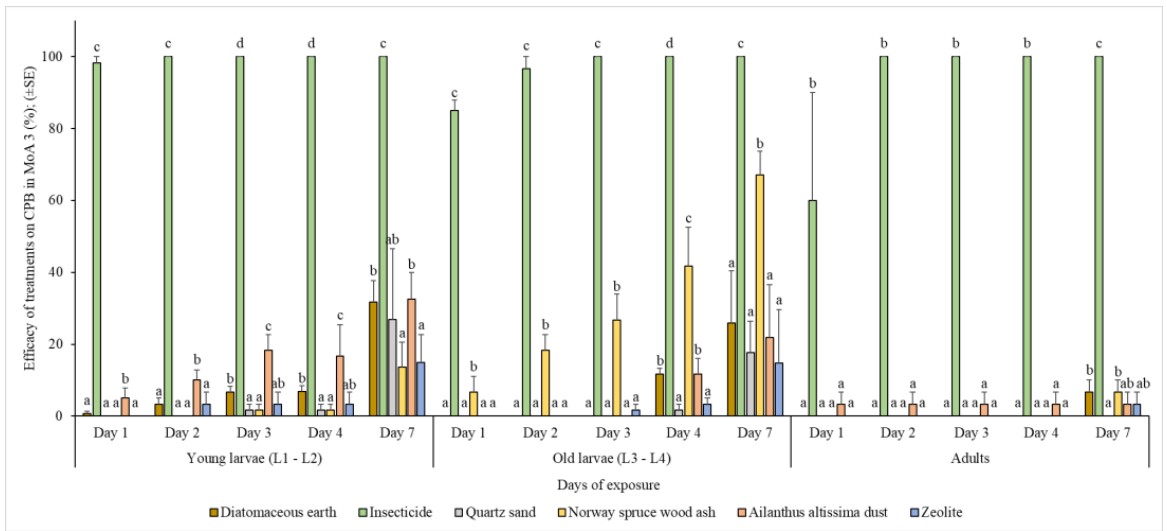

**Figure 3.** Efficacy of treatments (±SE) of CPB in MoA 3 (dusted beetles placed on undusted potato leaves) per day of exposure and developmental stage (different lowercase letters represent significant differences within the same developmental stage between different treatments by day of exposure).

Based on the analyses for the old larvae at the third MoA, there was a significant effect of the type of treatment (F = 194.16; df = 5; *p* < 0.0001) and the days of exposure (F = 18.12; df = 4; *p* < 0.0001) on the average efficacy. We also found that the interaction between the type of treatment and the days of exposure was not significant for old larval removal efficacy. From Figure 3, we can deduce that the most effective treatment was that of the positive control (100 ± 0.0%) after the second day of exposure. From the graph, we can also see that all of the studied inert dusts generated better results with this MoA. Wood ash was the most promising inert dust, with a 25% efficacy rate after 3 days of exposure (26.7 ± 7.3%). In regards to the last day (Day 7) of exposure, wood ash (67.0 ± 6,6%) was the most effective among the studied dusts, followed by diatomaceous earth (25.9 ± 14.4%), *A. altissima* dust (21.8 ± 14.8%), quartz sand (17.6 ± 8.8%), and zeolite (14.8 ± 14.1%).

For the adults at the third and final MoA, we found a significant effect of treatment (F = 208.01; df = 5; *p* < 0.0001) on the average efficacy, but no significant effect of days of exposure. We also did not find significant interactions between treatments and days of exposure. In regards to the treatment type, as shown in Figure 3, the overall efficacy of treat-

ments on the adult beetles was the highest for the positive control treatment (100 ± 0.0%). Between the studied inert dusts, there was little if no effect during the first 4 days of exposure. The only inert dust treatment that caused a very negligible effect in the first days of exposure was that of *A. altissima* leaf powder (3.3 ± 3.3). On the seventh day of exposure, the highest efficacy on the adult beetles between the used inert dusts was caused by wood ash (6.6 ± 3.3) and diatomaceous earth (6.6 ± 3.3).

*3.2. Defoliation*

3.2.1. General Insight

ANOVA of pooled results indicated a statistically significant effect of the day of exposure (F = 539.43; df = 4; $p < 0.0001$), type of treatment (F = 629.96; df = 6; $p < 0.0001$), mode of application (F = 181.57; df = 2; $p < 0.0001$), and the developmental stage of the CPB (F = 114.59; df = 2; $p < 0.0001$) on the average defoliation of the CPB in the experiment. The interactions were all statistically significant. The interaction between the day of exposure and the type of treatment (F = 20.59; df = 24; $p < 0.0001$), the day of exposure and the mode of application (F = 4.13; df = 8; $p < 0.0001$), the day of exposure and the developmental stage of the pest (F = 25.15; df = 8; $p < 0.0001$), the type of treatment and the mode of application (F = 22.46; df = 12; $p < 0.0001$), the type of treatment and the developmental stage (F = 20.17; df = 12; $p < 0.0001$), and, finally, the mode of application and the developmental stage of the pest (F = 18.13; df = 4; $p < 0.0001$). Other interactions that were statistically significant were between the day of exposure, the type of treatment and mode of application (F = 1.67; df = 48; $p < 0.0001$), the day of exposure, the type of treatment and the developmental stage of the pest (F = 2.92; df = 48; $p < 0.0001$), the day of exposure, the mode of application and the developmental stage of the pest (F = 2.12; df = 16; $p < 0.0001$), the type of treatment, the mode of application, and, lastly, the developmental stage of the pest (F = 12.17; df = 24; $p < 0.0001$). There was also a statistically significant interaction between the day of exposure, the type of treatment, the mode of application, and the developmental stage of the pest (F = 1.37; df = 96; $p < 0.0001$) on the average defoliation caused by the CPB in the experiment.

ANOVA of pooled results indicated a statistically significant effect of the days of exposure (F = 36.86; df = 3; $p < 0.0001$), the type of treatment (F = 168.42; df = 6; $p < 0.0001$) and the MoA (F = 22.65; df = 2; $p < 0.0001$) on the overall defoliation caused by the young larvae. Only two types of interactions affected the defoliation caused by the young larvae in the experiment. The interaction between the type of treatment and the days of exposure (F = 3.01; df = 18; $p < 0.0001$) and the interaction between the type of treatment and the MoA (F = 10.54; df = 12; $p < 0.0001$) were examined.

ANOVA of pooled results indicated a statistically significant effect of the days of exposure (F = 77.95; df = 4; $p < 0.0001$), the type of treatment (F = 246.49; df = 6; $p < 0.0001$), and the MoA (F = 42.12; df = 2; $p < 0.0001$) on the average defoliation caused by the old larvae in the laboratory trial. Only two types of interactions affected the defoliation caused by the old larvae in the experiment. The interaction between the type of treatment and the days of exposure (F = 5.20; df = 24; $p < 0.0001$) and the interaction between the type of treatment and the MoA (F = 13.37; df = 12; $p < 0.0001$) were examined.

ANOVA of pooled results indicated statistically significant effects of the day of exposure (F = 217.69; df = 4; $p < 0.0001$), the type of treatment (F = 286.87; df = 6; $p < 0.0001$), and the MoA (F = 187.68; df = 2; $p < 0.0001$) on the average defoliation caused by the adult specimen in the experiment. All three types of interactions affected the defoliation by the adult specimens of the CPB in the experiment. The interaction between the type of treatment and the days of exposure (F = 8.58; df = 24; $p < 0.0001$), between the day of exposure and MoA (F = 5.65; df = 8; $p < 0.0001$), and between the type of treatment and the MoA (F = 30.46; df = 12; $p < 0.0001$) were all examined.

3.2.2. Mode of Application 1

ANOVA of pooled results for the mode of application 1 indicated a statistically significant effect of the day of exposure (F = 107.69; df = 4; *p* < 0.0001), the type of treatment (F = 140.49; df = 6; *p* < 0.0001), and the developmental stage (F = 26.92; df = 2; *p* < 0.0001). All three affected the defoliation caused by the CPB. Some other interactions also had a statistically significant effect on the defoliation of the CPB in MoA 1. The interaction was between the day of exposure and the type of treatment (F = 4.89; df = 24; *p* < 0.0001), the day of exposure and the developmental stage of the pest (F = 4.08; df = 8; *p* < 0.0001), and the type of treatment and the developmental stage of the pest (F = 16.48; df = 12; *p* < 0.0001).

On the first day of exposure, we found, significantly, the lowest average defoliation rate caused by the CPB specimens (13.7 ± 2.8%). Grand means and parentheses are indicated as results, and are presented in this paragraph and the second paragraphs of the Sections 3.2.3 and 3.2.4 in this paper. The significantly highest average defoliation rate was recorded at the seventh day of exposure (69.9 ± 4.9%). Regarding the treatment, the highest average defoliation rate caused by the CPB specimens was recorded in the negative control treatment (83.9 ± 4.5%) and in the quartz sand treatment (83.6 ± 4.5%). Significantly, the lowest average defoliation was caused by the CPB specimens in the positive control treatment (0.6 ± 0.2%). Significantly, again, low average defoliation was also recorded in the *A. altissima* treatment (30.5 ± 4%). As for the developmental stage, the highest average significant defoliation among all developmental stages was recorded among the old larvae (57.5 ± 4.1%). Significantly, the lowest average defoliation among all of the developmental stages of the CPB was recorded among the adult beetles (40.8 ± 3.6%).

Based on the analysis, we found that at the first MoA, there was a significant effect of the type of treatment (F = 23.49; df = 6; *p* < 0.0001) and the days of exposure (F = 9.00; df = 3; *p* < 0.0001) on the overall defoliation caused by the young larvae. The interaction between them (F = 0.57; df = 18; *p* < 0.9028) was not significant, and it did not affect the overall defoliation caused by the young larvae. In regards to the treatment type, as shown in Figure 4, the overall defoliation by the young larvae on the third day of exposure was the highest for the negative control (100 ± 0.0%), quartz sand (95 ± 5%), and wood ash (86.7 ± 8.8%) treatments. The lowest defoliation rate was observed for the positive control treatment (0.0 ± 0.0). Among the studied inert dusts, the most promising results were obtained for zeolite and diatomaceous earth treatments, but, overall, the results did not differ much from those of the most ineffective treatments (neg. control, quartz sand, and wood ash). The lowest defoliation rate caused by the young larvae was recorded for the treatments of zeolite (71.0 ± 29.0%), diatomaceous earth (73.7 ± 26.3%), and *A. altissima* dust (88.3 ± 9.3%) on the seventh day of exposure (Figure 4).

Based on the analysis, we found that at the first MoA, there was a significant effect of the type of treatment (F = 89.89; df = 6; *p* < 0.0001) and the days of exposure (F = 26.99; df = 4; *p* < 0.0001) on the overall defoliation caused by the old larvae. The interaction between them (F = 2.03; df = 24; *p* < 0.0118) was not significant, and it did not affect the overall defoliation caused by the old larvae. In regards to the treatment type, as shown in Figure 4, the overall defoliation by the old larvae on the second day of exposure was the highest for the negative control (100 ± 0.0%), quartz sand (91.7 ± 4.4), and diatomaceous earth (80 ± 11.5%) treatments. The lowest defoliation was observed for the positive control treatment (0.67 ± 0.67%). Among the studied inert dusts, the most promising results we obtained for wood ash and *A. altissima* dust treatments, but, overall, they still had quite high defoliation percentages. The lowest defoliation rate caused by the old larvae was recorded for the wood ash (74.0 ± 20.7%) and *A. altissima* dust (32.7 ± 2.2%) treatments on the seventh day of exposure (Figure 4).

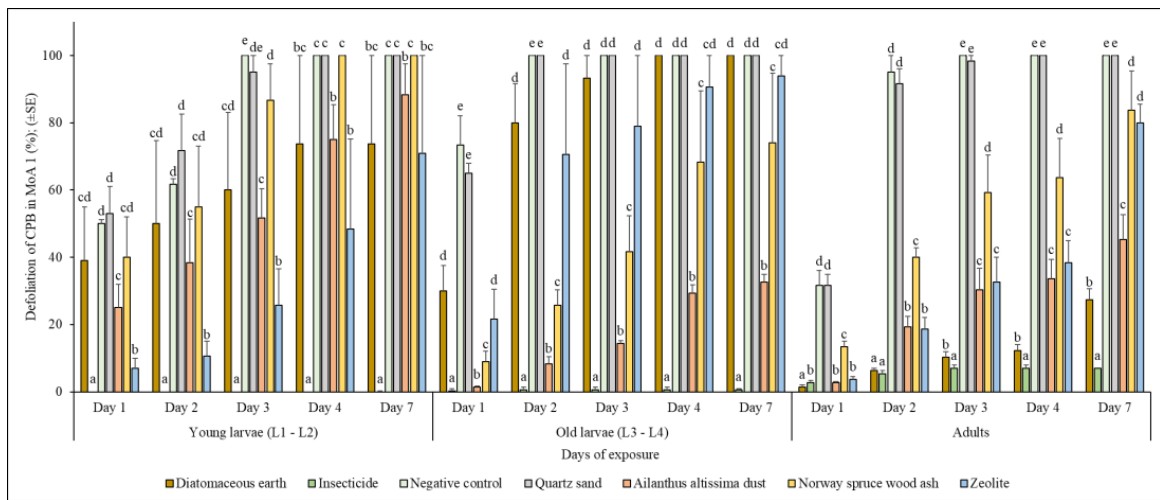

**Figure 4.** Defoliation (±SE) caused by the CPB in MoA 1 (dusted beetles placed on dusted potato leaves) per day of exposure and developmental stage in each treatment (different lower case letters represent significant differences within the same developmental stage between different treatments by day of exposure).

Based on the analysis, we found that at the first MoA, there was a significant effect of treatment (F = 234.55; df = 6; *p* < 0.0001) and day of exposure (F = 113.11; df = 4; *p* < 0.0001) on the overall defoliation caused by the adult specimen. The interaction between them (F = 8.48; df = 24; *p* < 0.0001) was also significant and affected the overall defoliation caused by the adults. In regards to the treatment type, as shown in Figure 2, the overall defoliation by the adult beetles in the negative control (95 ± 5.0%) and quartz sand treatment (91.7 ± 44%) peaked on the second day of exposure. Considering all of the days of exposure, the positive control treatment had the lowest defoliation rate (7.0 ± 1.0%). All studied inert dusts had little effect on defoliation during the entire 7-day treatment period. The lowest defoliation rate by the adult specimens was recorded in the treatments of diatomaceous earth (27.3 ± 3.3%) and *A. altissima* dust (45.3 ± 7.3%) on the seventh day of exposure (Figure 4).

### 3.2.3. Mode of Application 2

ANOVA of pooled results for the mode of application 2 indicated a statistically significant effect of the day of exposure (F = 207.96; df = 4; *p* < 0.0001), the type of treatment (F = 281.28; df = 6; *p* < 0.0001), and the developmental stage (F = 106.84; df = 2; *p* < 0.0001). All three affected the defoliation caused by the CPB. All interactions also had a statistically significant effect on the defoliation of the CPB in MoA 2. The interaction between the day of exposure and the type of treatment (F = 9.77; df = 24; *p* < 0.0001), the day of exposure and the developmental stage of the pest (F = 9.33; df = 8; *p* < 0.0001), the type of treatment and the developmental stage of the pest (F = 15.13; df = 12; *p* < 0.0001) and the interaction between all three of them, and, finally, between the day of exposure, the type of treatment, and the developmental stage of the pest (F = 2.34; df = 48; *p* < 0.0001).

On the first day of exposure, we found the lowest average defoliation rate caused by the CPB specimens (17.7 ± 3.3%). The highest average defoliation rate was recorded at the seventh day of exposure (74.7 ± 4.8%). Regarding the treatment, the highest average defoliation rate caused by the CPB specimens was recorded in the negative control treatment (87.9 ± 4.1%) and in the quartz sand treatment (86.2 ± 4.5%). Significantly the lowest average defoliation was caused by the CPB specimens in the positive control treatment (0.04 ± 0.03%). Significantly, low average defoliation was also recorded in the *A. altissima* treatment (31.5 ± 4.1%). As for the developmental stage, the highest average significant defoliation among all developmental stages was recorded among the old larvae (69 ± 3.8%).

Significantly, the lowest average defoliation among all of the developmental stages of the CPB was recorded among the adult beetles (43.9 ± 3.8%).

Based on the analysis, we found that in the second MoA, there was a significant effect of the type of treatment (F = 174.82; df = 6; *p* < 0.0001) and of the days of exposure (F = 22.98; df = 3; *p* < 0.0001) on the overall defoliation caused by the young larvae. The interaction between them (F = 3.16; df = 18; *p* < 0.0005) was not significant, and it did not affect the overall defoliation caused by the pests. Based on the results shown in Figure 5, we can also deduce that the overall effect of the used dusts was unsatisfactory. The most effective treatment was that of the positive control, with an 0% defoliation rate (0.0 ± 0.0%). Overall, the efficiency of the inert dusts was lower than we expected, except for the *A. altissima* dust treatment, which had only a 14.3% defoliation rate on the third day of exposure (14.3 ± 1.3%). On the seventh day of exposure, the lowest and most significant defoliation rate generated by the young larvae between the used inert dusts was also recorded for the *A. altissima* dust treatment (41.0 ± 4.2%) (Figure 5).

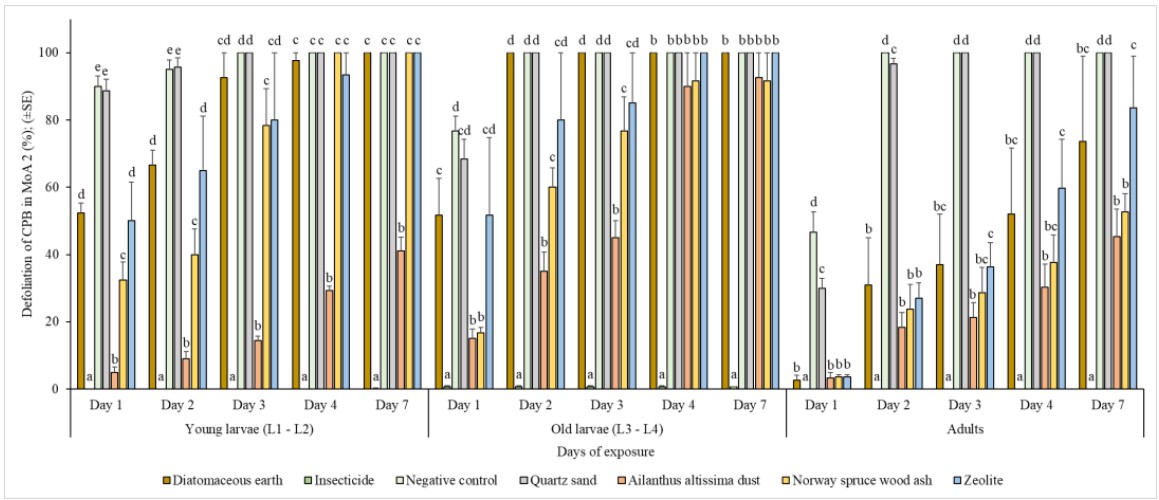

**Figure 5.** Defoliation (±SE) caused by the CPB in MoA 2 (undusted beetles placed on dusted potato leaves) per day of exposure and developmental stage in each treatment (different lower case letters represent significant differences within the same developmental stage between different treatments by day of exposure).

In the second MoA, we found a significant effect of treatment (F = 108.32; df = 6; *p* < 0.0001) and a significant effect of days of exposure (F = 42.10; df = 4; *p* < 0.0001) on the overall defoliation caused by the old larvae. We also found a significant interaction between the type of treatment and the days of exposure (F = 3.56; df = 24; *p* < 0.0001). From the graph (Figure 5), we can see that the overall efficiency of the used inert dusts was very low. By the fourth day of exposure, every treatment except that of the positive control had an overall defoliation percentage of more than 90%. The most effective treatment was that of the positive control, with less than 1% of recorded defoliation damage (0.67 ± 0.33%). The efficiency of the inert dusts was unsatisfactory.

For the adults in the second MoA, we found a significant effect of the type of treatment (F = 77.91; df = 6; *p* < 0.0001) and the days of exposure (F = 39.27; df = 4; *p* < 0.0001) on the overall defoliation. We also found a significant interaction between the type of treatment and the days of exposure (F = 2.60; df = 24; *p* < 0.0001). The results shown in Figure 5 indicate that the overall effect of the used dusts was somewhat similar to the previous MoA but less efficient. On the seventh day of exposure, the highest percentage of defoliation by the adult beetles between the used inert dusts was recorded for the treatments of quartz sand (100 ± 00%), the negative control (100 ± 0.0%), and zeolite (83.7 ± 15.3%). Better results were observed for the wood ash (52.7 ± 5.4%) and *A. altissima* dust (45.3 ± 8.2%) treatments. Overall, the most efficient treatment was the positive control (0.0 ± 0.0%).

### 3.2.4. Mode of Application 3

ANOVA of pooled results for the mode of application 3 indicated a statistically significant effect of the day of exposure (F = 311.96; df = 4; $p < 0.0001$), the type of treatment (F = 337.08; df = 6; $p < 0.0001$), and the developmental stage (F = 31.11; df = 2; $p < 0.0001$). All three affected the defoliation caused by the CPB. All other interactions also had a statistically significant effect on the defoliation of the CPB in MoA 3. The interaction between the day of exposure and the type of treatment (F = 12.40; df = 24; $p < 0.0001$), the day of exposure and the developmental stage of the pest (F = 22.62; df = 8; $p < 0.0001$), the type of treatment and the developmental stage of the pest (F = 10.94; df = 12; $p < 0.0001$) and the interaction between all three of them, and, finally, between the day of exposure, the type of treatment, and the developmental stage of the pest (F = 2.80; df = 48; $p < 0.0001$).

On the first day of exposure, we found the lowest average defoliation rate caused by the CPB specimens (25.1 $\pm$ 3.9%). The highest average defoliation rate was recorded at the seventh day of exposure (84.2 $\pm$ 4.6%). Regarding the treatment, the highest average defoliation rate caused by the CPB specimens was recorded in the negative control treatment (87.3 $\pm$ 4.2%) and in the quartz sand treatment (88.7 $\pm$ 4.2%). Significantly the lowest average defoliation was caused by the CPB specimens in the positive control treatment (0.6 $\pm$ 0.02%). Significantly, high average defoliation was also recorded in the diatomaceous earth treatment (81.9 $\pm$ 4.8%) and zeolite treatment (80.2 $\pm$ 5%). As for the developmental stage, the highest average significant defoliation among all developmental stages was recorded among the old larvae (74.8 $\pm$ 3.8%). Significantly the lowest average defoliation among all of the developmental stages of the CPB was recorded among the young larvae (62.6 $\pm$ 4.5%).

For the third and final MoA, there was a significant effect of the type of treatment (F = 260.32; df = 6; $p < 0.0001$) and the days of exposure (F = 35.67; df = 3; $p < 0.0001$) on the overall defoliation caused by the young larvae. The interaction between the type of treatment and the days of exposure was also significant (F = 11.26; df = 18; $p < 0.0001$). In regards to the treatment type, as shown in Figure 6, the overall defoliation by the young larvae for this third type of dust application was the highest compared to that of all other application methods. The positive control was the only treatment that had very good results after 7 days of exposure (0.0 $\pm$ 0.0%). For all of the studied inert dusts, there was little or no effect. On the third day of exposure, 100% defoliation by the young larvae was observed for all of the studied treatments (100.0 $\pm$ 0.0%), except the *A. altissima* dust treatment (45.0 $\pm$ 5.8%), which had at least some minimal effect on the defoliation by the young larvae (Figure 6).

In the third MoA, there was a significant effect of the type of treatment (F = 75.82; df = 6; $p < 0.0001$) and the days of exposure (F = 15.03; df = 4; $p < 0.0001$) on overall defoliation caused by the old larvae. The interaction between the type of treatment and the days of exposure (F = 1.70; df = 24; $p < 0.0442$) was not significant. In regards to the treatment type, as shown in Figure 4, the overall defoliation by the old larvae in this third type of dust application was the highest compared to that of all other application methods. The positive control was the only treatment that generated very good results after 7 days of exposure (0.67 $\pm$ 0.67%). For all of the studied inert dusts, there was little if no effect. On the third day of exposure, 100% defoliation by the old larvae was observed for all of the studied treatments (100 $\pm$ 0.0%), except the wood ash treatment (67.3 $\pm$ 22.2%), which had at least some minimal effect (Figure 6).

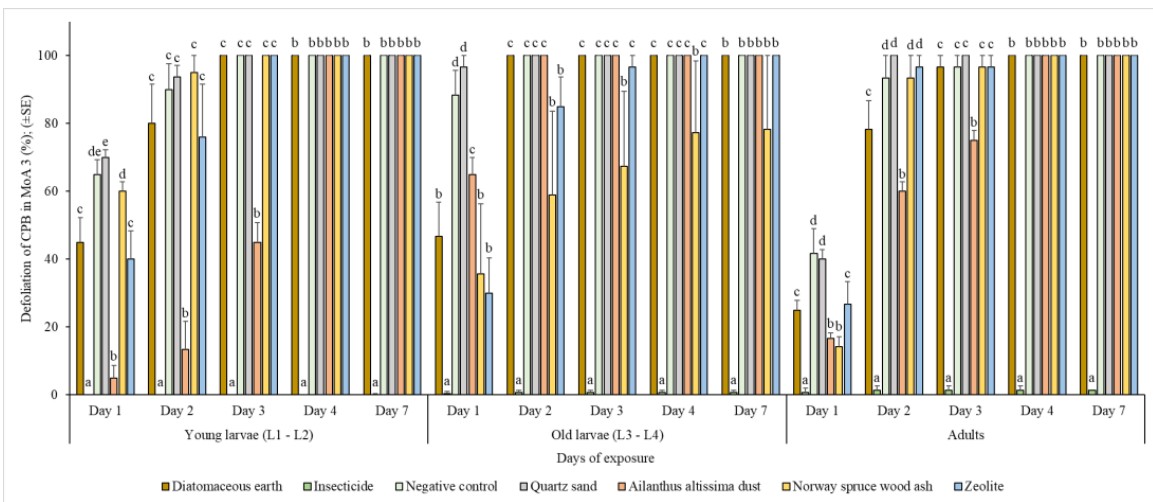

**Figure 6.** Defoliation (±SE) caused by the CPB in MoA 3 (dusted beetles placed on undusted potato leaves) per day of exposure and developmental stage in each treatment (different lower case letters represent significant differences within the same developmental stage between different treatments by day of exposure).

In the third MoA, we found a significant effect of treatment (F = 367.41; df = 6; $p < 0.0001$) and days of exposure (F = 375.82; df = 4; $p < 0.0001$) on the overall defoliation caused by the adults in the experiment. In general, we also found a significant interaction between the type of treatment and the days of exposure (F = 14.16; df = 24; $p < 0.0001$). In regards to the treatment type, as shown in Figure 6, the overall defoliation by the adult beetles in this third mode of dust application was the highest compared to all other MoA. The positive control was the only treatment that had a satisfactory effect after 7 days of exposure (1.3 ± 1.3%). For all of the studied inert dusts, there was little or no effect. On the fourth day of exposure, 100% defoliation by the adult beetles was observed for all of the studied treatments (100 ± 0.0%) (Figure 6).

## 4. Discussion

The results of the present research indicate that the type of inert dust, the days of exposure, and the MoA have an important influence on the efficacy and the overall defoliation of most studied developmental stages of the CPB. Regarding the days of exposure, we can conclude that at the end of the predetermined time period, the efficacy of treatments on all studied CPB developmental stages was higher than the efficacy of treatments in the first days of exposure. This applies to almost all treatments and modes of application. The exception was for the treatment of the positive control, which in some cases achieved an efficacy rate of 100% after the first day of exposure.

Regarding the MoAs, the results varied. Our results highlight the positive and negative properties of the MoAs and inert dusts studied in this research. Focusing on the mode of application, its efficacy varied among studied developmental stages of the CPB. The highest efficacy of the used inert dusts for young larvae was detected at MoA 1 and MoA 3. For the old larvae, the highest efficacy can be seen in MoA 3. The highest efficacy for adult beetles was similar in all three MoAs (unsatisfactory). We can conclude with certainty that the mode of application of the inert dust also affects the efficacy of the used dust on the pest when it is in the larval stage of its development. With both MoA 1 and MoA 3, the pest is directly exposed to the active substance, and therefore increases the chances of desiccation or cuticular damage [14,26], which most likely affects the higher efficacy in the aforementioned application modes 1 and 3.

One laboratory study showed that the adult and larval stages of the CPB are very sensitive to wood ash exposure over time [13]. We obtained similar results, as expected, but only for MoA of predusting of the pest (MoA 1 and MoA 3). High efficacy rates were not

achieved, but this may be because the specimens were dusted only once at the beginning of the trial. Usually, these kinds of results obtained in laboratory tests are better than those obtained in the open/field [2]. Boiteau et al. [13] showed that outdoor potato plants to which an ash barrier was added were just as damaged as the plants without an added ash barrier. The overall defoliation caused by the pest in our laboratory trials was also quite high: over 70% after 7 days of exposure in every MoA, and for every beetle developmental stage. Therefore, unmodified wood ash provided only limited CPB control in the laboratory and no control in the field. Some of the factors that could further contribute to the poor durability of the mentioned dusts on the applied surface may be exacerbated by various factors, including environmental factors such as the impact of wind and rain as well as agrotechnical factors such as irrigation and weed control. However, the efficacy of wood ash could be enhanced through the development of hydrophobic formulations, which is an area of research worth exploring [13]. From our geochemical analysis of the Norway spruce wood ash, we can see that it contained only 13.46% of $SiO_2$. In the study of Bohinc et al. [17], the most efficient types of wood ashes had the highest $SiO_2$ concentrations, yet as we can also see from our results, the value of $SiO_2$ in our wood ash in comparison with the percentage of $SiO_2$ in other used substances that were less efficient (quartz sand, zeolite) was almost negligible. For this reason, the higher effectiveness of Norway spruce ash should be linked not only to $SiO_2$ content but also to the content of its other chemical substances, which can independently or interactively affect the hygroscopic properties of these inert preparations [17,26,27]. As for the granulometric results, we could point out that the Norway spruce wood ash particles had the smallest grain size but were not as uniform as quartz sand or zeolite (Table 4). Batistič et al. [14], in their review on plant protection properties of inert dusts against CPB, also point out that wood ashes act as an obstacle for insects and physically upon contact. Wood ash damages the insect's epicuticular epidermis and its protective wax. It also acts hygrophilically, and, as a result, the insect loses water and dries out, i.e., desiccation of the insect occurs [13].

The other dusts tested were also quite ineffective. For diatomaceous earth, our results showed some effectiveness against young larvae (65.7 ± 22.9%; 31.7 ± 6.1%) and old larvae (25.9 ± 14.5%), but literally no effect against adults (6.7 ± 3.3%) after 7 days of exposure with MoA 1 or MoA 3, which were both pest pre-dusting MoAs. Defoliation was also very high for all modes of application and for all developmental stages of the pest, except for adult CPBs. With MoA 1, the adults only generated a 27.3 ± 3.3% defoliation rate after 7 days of exposure. The reason for this could be a higher efficacy of the used dust in MoA 1, but the overall efficacy between all three MoAs was still below the satisfactory level and can be considered ineffective against CPB adults. The research of Becker [12] presented the possibility of using different alternative methods to control the CPB on potatoes. She described the possibility of using diatomaceous earth as a potentially effective agent against the pest. For outdoor trials, they applied the substance in the form of a water mixture. In total, they applied it three times during the season. The final results were not impressive, as some of the other treatments achieved the same or even better results (e.g., attractive eggplant crop). Although diatomaceous earth and other silica-based dusts are known to be effective in controlling various storage pests by damaging their cuticle or by absorbing the protective wax layer that prevents body water loss, ultimately leading to the death of the insect death by desiccation [28], our laboratory trials using this substance were unsuccessful in effectively controlling the CPB. Hence, we can conclude that diatomaceous earth may not be an effective solution for managing all CPB developmental stages. We can also add that from our geochemical analysis, the diatomaceous earth that we used had a very low amount of $SiO_2$ (Table 2)—only 10.74% in comparison to other more effective, commercial preparations (SilicoSec®) containing higher $SiO_2$ percentages (>95%) [24]. In addition to the absorptive capacity and the insecticidal efficiency of this substance, diatomaceous earth is also influenced by the size of the particles, the uniformity and shape of the particles, the pH, and the purity of the formulation [11]. Baliota and Athanassiou [29] describe that modifications such as finer sieving, drying, and smashing diatomaceous earth particles

would improve their insecticidal values. Therefore, smaller particles could be more efficient than the large ones. This can explain the poorer efficiency of our tested diatomaceous earth and its granulometric results (Table 4).

Based on our data, it is clear that *A. altissima* leaf dust did not show a significant effect on the overall efficacy of any of the studied developmental stages of the pest. The highest efficacy of *A. altissima* was achieved for the young larvae with the third MoA (32.4 ± 7.4%). Regarding the overall defoliation, there was a surprisingly low percentage of defoliation with MoA 1 and MoA 2 for all developmental stages (young larvae [41.0 ± 4.3%]; old larvae [32.7 ± 2.2%]; adult beetles [45.3 ± 45.3%]), which are both modes of pre-dusting leaves (the leaves were covered in *A. altissima* leaf dust). A possible explanation for the low defoliation rate despite the low efficacy generated by *A. altissima* leaf dust treatment could be that it has an antifeedant effect on the pests. Boukhlika et al. [30] studied tomato plant responses to attack by the tomato borer (*Tuta absoluta* [Meyrick]). They found that some phenolic compounds, including catechin hydrate, which was also present in quite high concentrations in our *A. altissima* samples (Table 3), were de novo biosynthesized due to the attack of the borer. Andonova et al.'s [31] study provides us with the precise phenolic composition of *A. altissima*, from which we can infer that catechin hydrate is also detected in their samples. Catechin hydrate was also identified as a deterrent of the European corn borer (*Ostrinia nubilalis* [Hübner]) [32]. Therefore, this substance is most likely a phytoalexin, and its antifeedant effect could explain the lower damage to our potato leaf samples by the CPB, which is why it should be studied more thoroughly. Other studies have also analysed the effect of different plant extracts and powders on the CPB. Przybylski [33] notes a repellent effect of *Tanacetum vulgare* (L.) used as a dust and in the form of extracts on all of the developmental stages of the CPB. Plant extracts of *Angelica archangelica* (L.), *Grindelia camporum* (Greene), and *Inula auriculata* (Boiss and Balansa) show a very high level of feeding inhibition (antifeedant effect) of CPB larvae [34]. Among the potential plant extracts that could inhibit the development of larvae and adults of the CPB, Scott et al. [35] listed extracts from plant species of the Piperaceae family. Lu and Wu [36] tested the contact and fumigant efficacy of an extract prepared from the bark of the tree of heaven. Overall, their results were good, and the mortality of insects tested with the contact application (72 h for 70% mortality) was slightly lower than that in the form of a fumigant (24 h for 80% mortality). Mastelić and Jerković [37] studied the volatile constituents from the leaves of young and old *A. altissima* trees. They found sesquiterpenes, which are known to have some insecticidal activity. Some studies have shown potential for the use of plant extracts for CPB control, but new ideas should be applied in ongoing studies to better select substances that can be used economically and effectively against this harmful pest. Our results for *A. altissima* leaf dust did not show great efficacy but they did show an effect on their feeding ability. This is precisely why we believe that the mentioned substance and its compounds have potential in plant protection applications.

The next substance that we studied and would like to describe more precisely is zeolite. Our data clearly showed that zeolite did not have a significant efficacy on the CPB. The highest efficacy of zeolite was observed for young larvae in the first MoA (46.3 ± 27.3%). Regarding defoliation, it exceeded 70% in all developmental stages of the pest with all modes of application after 7 days of exposure, which is significantly higher than that observed for other more effective treatments. Therefore, we conclude that this type of inert dust appears to be ineffective against the CPB in our laboratory trials. Other researchers have confirmed that the use of zeolite as an inert dust against storage pests such as *Sitophilus zeamais* (Motschulsky) or *Sitophilus oryzae* (L.) could serve as a method of storage pest control [38–40]. Although their results seem to present zeolite as an efficient substance in storage pest control, it cannot be considered a substance that effectively suppresses the CPB outdoors, since the laboratory results obtained thus far are quite unsatisfactory. We could also point out that the grain size of zeolite is not as uniform as that of quartz sand or Norway spruce ash (Table 4), and that it is also a fairly dense sub-

stance (Table 1). Its $SiO_2$ content is also high (Table 2), but it still has not shown promising properties for CPB control.

The last of the studied inert dusts was quartz sand. We conclude that quartz sand did not show any significant efficacy on the studied developmental stages of the pest despite having the highest amount of $SiO_2$ (Table 2) based on our geochemical analysis. Its efficacy was lower than that recorded for some of the other treatments (wood ash, diatomaceous earth, etc.). The highest efficacy that was recorded for the quartz sand treatment was obtained on the 7th day of exposure using the third MoA for young larvae ($26.9 \pm 19.6\%$) and using the first MoA for old larvae ($25.3 \pm 9.0\%$). The overall efficacy of quartz sand treatment on adult CPBs was even lower ($13.3 \pm 8.8\%$). The same applies for the overall defoliation. In all of the studied modes of application (MoA 1, MoA 2 and MoA 3) and for every developmental stage of the pest, after the second day of exposure, defoliation percentages exceeded 90%, which is another indicator of the ineffectiveness of this substance. Other research studies have shown similar results for work with storage pests and quartz sand. Rojht et al. [41] studied the insecticidal efficiency of five different types of quartz sand obtained at different locations in Slovenia. The results of the research showed that the samples used had only a slight insecticidal effect on adult individuals of the rice beetle, and are therefore not suitable for wider use against storage pests. Some of the reasons that might have led to a lower overall efficacy in our trial might be connected with the overall particle size (Table 4). The particles of quartz sand used in our trial are thus quite large, but finer grinding of this substance is not recommended in practice, as it has negative effects on people and can cause silicosis on the lungs in case of excessive exposure [42]. Therefore, we used this predetermined size of the particles. One of the reasons for not using finer dust particles was also the difficulty of crushing or grinding them into a smaller/finer powder, which would most likely have a better effect on the pest [43].

## 5. Conclusions

This study investigated the effectiveness of different modes of application (MoA) and inert dusts on controlling Colorado potato beetles (CPB) at various developmental stages. The results showed that efficacy varied depending on the developmental stage, with MoA 1 and 3 being more effective against young larvae and MoA 3 being more effective against old larvae. Wood ash and diatomaceous earth were found to be the most promising inert dusts for causing significant efficacy among the studied treatments, while the efficacy on adult CPBs was still very low in all treatments. Geochemical analysis showed that wood ash and diatomaceous earth had low $SiO_2$ percentages, which was surprising considering their effectiveness in controlling the CPB. Other inert dusts (zeolite, quartz sand) had higher $SiO_2$ percentages but were less effective against CPB. Wood ash has a smaller particle size and acts hygrophilically, while diatomaceous earth destroys the insect's cuticle and/or absorbs the protective wax layer, contributing to their superior efficacy. The studied inert dusts were mostly ineffective in reducing CPB defoliation, except for *A. altissima* leaf dust treatment, which had some success, possibly due to the presence of Catechyin hydrate, which was confirmed as most abundant polyphenol by our chemical analysis of the plant material, and it has been previously shown to influence the antifeedant behavior of other pests.

Future research should focus on wood ash, diatomaceous earth, and *A. altissima* leaf dust as they show the greatest potential in controlling the CPB. Wood ash and diatomaceous earth were found by other research to be ineffective in controlling CPB in the field, despite limited efficacy in our laboratory studies. However, there is potential to improve their effectiveness through the development of hydrophobic formulations. Application methods should also be considered, such as electrostatic sprayers, and particle size may also play a role in their effectiveness. For *A. altissima*, we can point out that catechyin hydrate shows potential as an important alternative in CPB control, especially in terms of its use and tests as an antifeedant substance. Future prospects may also include studying the potential

synergism of the most effective substances in our research. Combining different substances or adding new ones, including biotic agents, may lead to greater efficiency.

**Author Contributions:** L.B., Methodology, Investigation, Writing—Original Draft Preparation; T.B. Methodology, Formal Analysis; A.H. Methodology, Data Curation; I.J.K. Methodology, Data Curation; S.T. Conceptualization, Methodology, Resources, Writing—Review and Editing, Supervision, Project Administration, Funding Acquisition. All authors have read and agreed to the published version of the manuscript.

**Funding:** The paper was written as part of the L4-3178 applied research project, which is funded by the Slovenian Research Agency (ARRS) and the Ministry of Agriculture, Forestry and Food of the Republic of Slovenia.

**Institutional Review Board Statement:** Not applicable. Study does not involve humans or animals.

**Informed Consent Statement:** Not applicable. Study does not involve humans or animals.

**Data Availability Statement:** The data presented in this study are available on request from the corresponding author.

**Acknowledgments:** The authors acknowledge the support of the Slovenian Research Agency (ARRS) within the infrastructural centre IC RRC-AG (IO-0022-0481-001).

**Conflicts of Interest:** The authors declare no conflict of interest. The funders had no role in the design of the study; in the collection, analyses or interpretation of the data; in the writing of the manuscript; or in the decision to publish the results.

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
