# Peer review of "Laboratory Investigation of Five Inert Dusts of Local Origin as Insecticides against the Colorado Potato Beetle (Leptinotarsa decemlineata [Say])"

_agronomy, doi:10.3390/agronomy13041165_

Round 1

Reviewer 1 Report (Previous Reviewer 2)

Dear authors,

You significantly improved the manuscript. However, I found some additional minor mistakes and I suggest to correct them.

Kind regards

Author Response

Reviewer 2 Report (Previous Reviewer 1)

The author answered most of my questions effectively. I think it can be accepted after a few minor modifications.

Line 16: Remove the space in front of The.

Line 22: Ailanthus altissima should be A. altissima.

Lines 18 & 19: There should be a space between the numerical and unit symbols. Need full text check.

Table 1, 2 and 3: Are these values averages? If yes, it is necessary to express as means ± SEs.

Line 223: P values in italics.

The figure is still difficult to distinguish, it is suggested to make the picture beautiful again, so that the content of each column can be clearly distinguished.

The contents of discussion and conclusion are different. The results of this paper should be discussed in detail in combination with references, and the conclusion should be summative. It is suggested that the author rewrite the discussion and conclusion, and the references in the conclusion and their descriptions be moved to the discussion.

Author Response

Reviewer 3 Report (New Reviewer)

ABSTRACT

-Line 16: Explain which application methods you applied.

-Line 18: “the first MoA” appear for the first time. Abbreviation MoA should be explained as well as “the first”. Please, explain here or in line 16.

-Line 20: “under different application methods”. This is not clear. You mentioned single number for mortality and defoliation. To which method are these numbers related?

-Lines 20-23: Provide percentages as for adults and old larvae.

INTRODUCTION

-Line 37: Delete “if it occurs”.

-Line 52: Is this correct? Which conventional insecticide act as physical poison?

-Lines 55-56: Replace “as the result of” with “due to”.

-Lines 63 and 66: Commas can be omitted.

-Line 72: Delete “etc.” and add 1-2 additional references.

-Line 75: Replace “our studied pest” with “CPB”.

-Lines 78-80: “through what mechanisms” is not clear. Also, you should mentioned earlier in the sentence that you performed laboratory trials.

MATERIAL AND METHODS

-Line 97: Add “untreated” before “fresh”.

-Lines 106-107: Did you use equal number of beetles/larvae from the first and the second generation? Possibly, generations may differ in sensitivity to control agent.

-Lines 114-116: It would be nice if you explain which information you get by these different methods or, possibly, which realistic situation you mimic by different MoA.  

-Lines 121-122: Provide volume of small containers. How do you maintain freshness of leaves?

-Line 124: Did you put equal number ofadult females and males in container?

-Line 152: “or days after exposure” is not necessary.

-You did not describe how you monitored defoliation.

-Table 3- Two columns are two measurements of the same sample? Why don’t you present average value and SE of the two measurements?

-Table 4: Correct 196,43.

-Line 213: How did you correct defoliation with Abbott’s formula.

RESULTS

-MAJOR REMARKS:

1. Experimental groups with zero variances should not be involved in ANOVA model.

2. “Day” is repeated measure factor and thus repeated measures ANOVA should be applied. However, in mortality data, you can do it only for young larvae from the 3rd to 7th day after you exclude positive control.

3. Old larvae and adults – you can compare groups with non-zero variances by 1-way ANOVA.

4. It is difficult to follow such thin bars. I suggest you to make 3 subfigures (A,B,C) for each developmental stage and arrange them one below other (not in the same line). Positive control should be the first bar.

5. Defoliation: Many experimental groups have 100% defoliation or, in the case of positive control, 0%. In these groups variance is equal to zero. Remarks are the same as for mortality data. Please, chose groups where you can apply multifactor ANOVA (e.g., initial days).

6. Defoliation- Why data on defoliation by young larvae on the first day are missing?

RESULTS

OTHER REMARKS:

-Line 232: Start with “ANOVA for young larvae…”.

-Lines 260-269: Are values in parenthese grand means? “29,1” in line 262 should be replaced with “29.1”. Check the whole text for such mistakes.

-FIGUREs 1, 2, 3: Explain MoA 1, 2, 3 in parentheses and in line 299,349,406 put “…(different lowercase letters represent significant differences……)”

-Lines 308-310: What does it mean “significantly the lowest” and “significantly highest”? It is significantly higher/lower comparing to other groups? Please, explain better here and in other parts of the text.

-Lines 322 and 333-334: “was not the best” is not good expression.

-Line 343: Explain what is similar.

-Line 344: “highest significant ”. You mean “significantly higher than….”.

-Lines 354-355: Delete this sentence and add “on the mortality” at the end of the previous sentence.

-Lines 362 and 365 and 509: Correct 32,1 and 1,7 and 3,3.

-Line 370: Mention in prentheses that third MoA refer to dusted CPB on untreated leaves.

-FIGURES 4, 5, 6: Explain MoA 1, 2, 3 in parentheses and in line 513,572,632 put “…(different lowercase letters represent significant differences……)”

-Line 519: Delete “other”. In previous sentence you mention only main effects.

DISCUSSION

-Lines 637-640: Replace “treatment” with “inert dust” and deleate the end of the sentence “in association with ….”.

-Lines 646-648: Give some suggestions/explanations on different outcomes of the three MoA. Are there similar studies that examine different MoA with inert dusts? You provided some explanations in lines 774-783. Plese, consider to move that part of the text here.

-Lines 654-656: Why are inert dusts inefficient in the field? Give possible explanations.

-Line 670: Delete “study” and add something like “on plant protection properties of inert dusts against CPB”.

-Line 678: Add “pest” before “predusting”.

-Line 679: Replace “one” with “adults” or the end of sentence can be “….for larvae but not for adults.”

-Line 681: Mortality was not high in adults. So, there must be some other explanation.

-Lines 687-691: Maybe, cuticle was not destroyed due to low attachement of particles. You do not have data on damages of cuticle and water loss. Please, reformulate the sentence.

-Composition of Ailanthus altissima: Compare your results with results of other studies. For example, Mastelić and Jerković (2002, Croatica chemica acta, 75, 189-197) found sesquiterpenes. For phenolic compounds you can cite Andonova et al. (2023, Plants, 12, 920).

-Line 773: Delete “5 Conclusions”. The rest of the text is continuation of the Discussion.

-Line 774: Delete “In conclusion”.

-Lines 788-796: Give some suggestions to explain differences among developmental stages. Is it body size, cuticle composition? Diifferences among CPB developmental stages have been also shown for other possible means of control and you can cite some papers on that subject.

-Line 802: “efficient efficiency” is strange.

-Paragraph that starts with line 797: It seems that you already discussed characteristic of inert dusts. Does this part refer to effects on defoliation? Please, mention that to be more clear.

-Line 831: Replace “etc.” with some additional references.

Round 2

Reviewer 3 Report (New Reviewer)

I am satisfied with the majority of manuscript corrections. Responses to my comments and suggestions are well presented and it was easy to follow manuscript corrections. However, I still have one major remark and one minor suggestions.

Minor suggestion regarding grand means: I agree with your presentation of general (pooled) analyses. However, I was not sure that you put grand means  and parentheses. Can you explain that in the text at several places? This is only a suggestion.

Major remark regarding statistical analyses: I will accept ANOVAs that include groups with zero variances. I sow that some highly ranking journals accept such analyses. To me it is not correct but I will not insist further. However, repeated measures ANOVA must be performed in analyses where day is one of the factors. The difference between ANOVA and repeated measures ANOVA is similar to difference between t-test for independent samples and t-test for dependent samples. You monitored mortality or defoliation during time in each replicate. Accordingly, you cannot compare, for example, all replicates on day 1 with all replicates on day 2 because mortality in replicate 1 on day 1 is connected with mortality in replicate 1 on day 2 .................replicate 3 on day 1 is connected with mortality in replicate 3 on day 2.

In the study of Lopez-Garcia et al. (2018, Testing the insecticidal activity of nanostructured alumina on Sitophilus oryzae (L.)(Coleoptera: Curculionidae) under laboratory conditions using galvanized steel containers. Insects, 9,3, 87) you can find the sentence "In the mortality assessment, exposure time was the repeated measure variable."

Author Response

This manuscript is a resubmission of an earlier submission. The following is a list of the peer review reports and author responses from that submission.

Round 1

Reviewer 1 Report

This manuscript is a preliminary study of the use of five inert dusts to control CPB, it provides a alternative control methods for CPB. The subject matter had certain research meaning but there are major problems with the manuscript that prevent me from recommending it in its current form for publication.

1. The results were poorly described. The author divided the biotest results of insects at three different developmental stages (Adults, Old larvae and Young larvae) into three parts, and listed the results of two indicators of mortality and defoliation respectively at each developmental stage. This makes it difficult to compare CPB at different stages of development. The current description was too tedious and repetitive. It is suggested that the results of the three developmental periods be put together, and that the overall results be divided into two parts, one is mortality and one is defoliation. The description of results should focus on comparing the biotest results from three developmental stages, rather than repeating the description of each one.

2. The figures were badly made. First, because the mortality rate of insecticide treatment group is particularly high, it is suggested to truncate the ordinate, so as to show the differences of other treatments more directly. Second, why are there no negative control data in mortality outcomes? Third, why were the results of defoliation not tested for variance?

3. Will the application of such inert dusts affect the growth, development and yield of tomatoes? Whether the author has done the relevant measurement.

4. The author made a detailed biogenic test of five kinds of inert dusts, and also made a detailed analysis of the dust composition. Should you choose the dust that can effectively cause the death of beetles, and specifically identify which components of the dust are playing the main role? Otherwise the present results do not have any in-depth mechanism of study.

5. Line 40: possibly due to an antifeedant effect on the beetles this result is not mentioned in the experiment.

6. Line 95: It is suggested to describe the experimental ideas according to the reported results so as to compare with previous studies and understand the main innovations of this paper.

7. Line 112: Why did the author choose spinosad for the positive control in this experiments?

8. The description of Lines 241-243 is repeated with Lines 253-254 and 258-260.

9. Lines 468-470: “The most effective treatment was that of the positive control, with a 100% mortality rate on the second day of exposure (100.0±0.0%) and therefore a 0% defoliation rate (0.0±0.0%).” This sentence does not correspond to the figure.

10. Lines 501-503: “Regarding the MoA, the results varied. The highest mortality was recorded for the first MoA, followed by the third and then the second MoA in predominantly all of the studied CPB developmental stages.” The highest mortality was recorded for the third MoA in Old larvae (L3-L4), how to explain this difference ?

11. It is not recommended to introduce new literature points to discuss in the conclusion part.

Author Response

Detailed answers to the reviewer's questions/comments are available in the attachment.

Reviewer 2 Report

Dear authors

your study presents very interesting results on the efficacy of different inert dusts on CPB. Although most of the dusts studied were not very effective, there are some promising results with wood ash, which showed limited control, and Ailanthus altissima leaf dust, which reduced defoliation. Both the positive and negative results are worth publishing to gain a better understanding of potential compounds that can be used as alternatives to chemical pesticides for CPB control.

I have encountered some issues that need to be clarified before publishing your work.

L135: The volume of the container is larger than 1 L (if you calculate the volume based on the dimensions)

L156: Please describe the method of application of spinosad in all three MoAs.

L186: Please rearrange the references

Table 3: The heading of the columns is missing

L212: You mention a laser granulometer here, is it the same as described in line 205? Please add information about the manufacturer.

Kind regards

Author Response

(The authors gave the same response as above.)
